# Temperature, species identity and morphological traits predict carbonate excretion and mineralogy in tropical reef fishes

**Mattia Ghilardi** [1,2] ✉, **Michael A. Salter** [3], **Valeriano Parravicini**[4,5], **Sebastian C. A. Ferse** [1,2], **Tim Rixen** [1], **Christian Wild**[2], **Matthias Birkicht**[1], **Chris T. Perry** [6], **Alex Berry**[3], **Rod W. Wilson** [3], **David Mouillot** [7,5] & **Sonia Bejarano** [1]

Anthropogenic pressures are restructuring coral reefs globally. Sound predictions of the expected changes in key reef functions require adequate knowledge of their drivers. Here we investigate the determinants of a poorly-studied yet relevant biogeochemical function sustained by marine bony fishes: the excretion of intestinal carbonates. Compiling carbonate excretion rates and mineralogical composition from 382 individual coral reef fishes (85 species and 35 families), we identify the environmental factors and fish traits that predict them. We find that body mass and relative intestinal length (RIL) are the strongest predictors of carbonate excretion. Larger fishes and those with longer intestines excrete disproportionately less carbonate per unit mass than smaller fishes and those with shorter intestines. The mineralogical composition of excreted carbonates is highly conserved within families, but also controlled by RIL and temperature. These results fundamentally advance our understanding of the role of fishes in inorganic carbon cycling and how this contribution will change as community composition shifts under increasing anthropogenic pressures.

Ecosystems globally are rapidly restructuring into novel configurations in response to anthropogenic pressures[1,2]. These profound changes have wide-reaching implications for ecosystem functioning, climate warming mitigation, the provision of ecosystem services, and human wellbeing[3–7]. To better understand, anticipate, and address the impact of these changes on ecosystems, a detailed understanding of key ecosystem functions and their drivers is critical.

In addition to being ecologically, nutritionally, and economically important, fishes are major contributors to biogeochemical cycles in the global ocean[8–13]. Beyond contributing to nutrient cycling[9,13–16], marine bony fishes (Teleostei) significantly influence the biological carbon pump[12] and the cycling of inorganic carbon through the excretion of fine-grained carbonates as a by-product of osmoregulation[8,17,18]. Fish precipitate ingested calcium and magnesium ions as carbonate crystals within their alkaline and bicarbonate-rich intestinal fluid and excrete them at high rates either within mucus-coated pellets or in faeces[17–20]. This process reduces the osmotic pressure in the intestinal fluid, thus facilitating water absorption into the blood[18,21]. It also plays an important role in the fish's calcium homeostasis[22], protecting them against renal stone formation by preventing intestinal calcium being absorbed into the blood[23]. Fish carbonate excretion is estimated to potentially represent ~15% (8.9 Tmol year⁻¹) of the global carbonate production in surface oceans, with less conservative estimates as high as 45%[8].

¹Leibniz Centre for Tropical Marine Research (ZMT), Fahrenheitstraße 6, 28359 Bremen, Germany. ²Department of Marine Ecology, Faculty of Biology and Chemistry, University of Bremen, Leobener Straße UFT, 28359 Bremen, Germany. ³Biosciences, University of Exeter, Exeter EX4 4QD, UK. ⁴PSL Université Paris: EPHE-UPVD-CNRS, USR3278 CRIOBE, University of Perpignan, 66860 Perpignan, France. ⁵Institut Universitaire de France, Paris, France. ⁶Geography, University of Exeter, Exeter EX4 4RJ, UK. ⁷MARBEC, Univ Montpellier, CNRS, Ifremer, IRD, 34095 Montpellier, France. ✉e-mail: mattia.ghilardi91@gmail.com

The rate of carbonate excretion by fish is assumed to be proportional to metabolic rate since it is directly related to drinking rate and thus to the amount of calcium and magnesium ingested through seawater, which determines carbonate excretion rate[18,24,25]. Although this assumption remains untested, fish contribution to global oceanic carbonate production has been estimated by combining metabolic theory[26] with observations of carbonate excretion rates for two benthic species[8]. Consistent with metabolic theory, carbonate excretion rate was later found to decrease disproportionately (i.e., to scale hypoallometrically) with body mass across reef fishes[27,28], and to be positively related to water temperature in the sheepshead minnow (*Cyprinodon variegatus*)[8] and Gulf toadfish (*Opsanus beta*)[29]. Although these results suggest a direct link between carbonate excretion rate and metabolic rate, a rigorous investigation is needed to confirm the consistency of this relationship because of its potential key influence on large-scale carbonate production. Furthermore, several other environmental factors (e.g. salinity, $CO_2$) and fish traits (e.g. activity, diet, intestinal length) are known or expected to influence carbonate excretion directly or indirectly[12,25,30,31]. However, a comprehensive analysis of the factors determining interspecific differences in fish carbonate excretion rate is lacking, but needed to refine assessments of large-scale carbonate production. Such analysis will be crucial to scale carbonate excretion up from the individual to the community level, thereby increasing our understanding of the contribution of fishes to global carbon cycling in the ocean.

Fish excrete carbonate at high rates (i.e., at least up to 105 g m$^{-2}$ yr$^{-1}$ on coral reefs)[28]. The global significance of this process lies in the typically high Mg/Ca ratios and low degrees of crystallinity in the excreted carbonates compared with most other biogenic marine carbonates[8,17,27,28,32,33], which implies relatively high solubility[34]. Therefore, fish carbonates are hypothesised to be an important source of upper ocean carbonate dissolution, which is predicted to occur based on observed alkalinity–depth profiles, but for which sources remain enigmatic[8,35]. Mesopelagic fish, the largest biomass of fish (and vertebrates) on the planet, may for instance drive an upward alkalinity pump by producing carbonates at depth and excreting them at the peak of their vertical feeding migrations, where they dissolve rapidly releasing new alkalinity to the surface ocean[36]. However, fish are known to produce a wide variety of carbonate polymorphs, including low- and high-magnesium calcite (LMC and HMC, respectively), aragonite, monohydrocalcite (MHC), and amorphous calcium magnesium carbonate (ACMC)[27,28,32,33,37,38], with respective solubilities spanning several orders of magnitude if existing solubility data from mostly non-fish sources are applied[34,39–41]. The mineralogical composition of excreted carbonates has been characterised for a wide range of tropical, subtropical, and temperate fishes, showing a high degree of consistency within families, with a few exceptions[28,32,37,38]. This strong taxonomic conservatism allowed for the first estimates of polymorph-specific production rates by combining individual carbonate excretion rates with family-average carbonate composition[28,37]. Important regional differences in the preservation potential of fish carbonates, driven by variation in fish community composition, were highlighted[28]. However, despite the apparent family-level consistency, further determinants of the mineralogical composition of fish carbonates have yet to be investigated. Identifying the factors that govern fish carbonate mineralogy and incorporating carbonate composition within production models is essential for assessing (1) the current contribution of fishes to both open ocean and shallow marine carbonate budgets and (2) the impacts of ongoing fishing- and climate-induced changes in fish community composition on ecosystem functioning.

Here, we aimed to identify the environmental factors and fish traits that best explain variation in the excretion rates and mineralogical composition of excreted carbonates. We focus on tropical and subtropical reef fishes as they represent most of

marine vertebrate biodiversity within a small fraction of the ocean[42]. We assembled the largest database available to date, including carbonate excretion rates from 382 individuals across 85 fish species and 35 families, spanning a wide range in body mass (<1 g to >10 kg) and trophic level (2.0–4.5). Data were collected in three tropical and subtropical regions (180 individuals from 29 species in Australia, 90 individuals from 10 species in the Bahamas, and 112 individuals from 46 species in Palau) within three marine biogeographic realms[43]. Retaining only families with at least three independent observations (352 individuals from 71 species and 21 families), we tested whether fish traits (body mass, caudal fin aspect ratio−AR [a proxy for general activity level[44]], relative intestinal length−RIL [intestinal length relative to body standard length]), environmental variables (temperature, salinity), and taxonomic identity (i.e., family) are significant predictors of carbonate excretion rate, and whether the latter is influenced by the duration of the total sampling period. Then, we assessed whether the predictors identified in the previous step (i.e., all excluding salinity and total sampling period) can accurately predict the excretion rate of five major carbonate polymorphs produced by fishes (i.e., LMC, HMC, aragonite, MHC, ACMC), and whether temperature, RIL, and taxonomic identity influence carbonate mineralogy. We confirm that carbonate excretion rate scales proportionally with metabolic rate through the effect of body mass, temperature, and AR, and show that, per unit mass, it decreases disproportionately with body mass and RIL. This implies major changes in community-level carbonate production with ongoing human- and climate-induced shifts in fish size and trophic structure on global reefs[6,45]. One step further, we show that carbonate mineralogy is highly taxonomically conserved but also controlled by RIL and water temperature, providing a fundamental advance in quantifying fish contribution to carbonate budgets.

## Results

### Predictors of total carbonate excretion

A Bayesian multilevel distributional regression model exploring the relationship between total carbonate excretion rate and body mass, AR, RIL, temperature, and family explained 85.5% (95% credible interval (CI): 83.7%, 86.8%) of the variation in the response and showed a strong relationship between observed and predicted excretion rate (Supplementary Fig. 1), with 96% of the observed values falling within the 95% CIs of the predictions.

Fish body mass was the strongest predictor of carbonate excretion rate. RIL had a stronger influence on carbonate excretion rates compared to temperature and AR, which had the weakest effect (Fig. 1a). Further, taxonomic identity explained a minor proportion of variance in the dataset (∼5%), with a few families (Labridae excluding Scarini, Lutjanidae, Pomacentridae, and Terapontidae) clearly deviating from the average estimate (Fig. 1b).

There was a positive relationship between excretion rate and all three factors related to metabolic rate: body mass, temperature, and AR (Figs. 1a and 2a–c). Small fishes excreted more carbonate per unit mass than large ones, as indicated by the hypoallometric relationship between excretion rate and body mass (both natural-log transformed; mean and 95% CI: $\beta = 0.78$ [0.72, 0.83]) (Fig. 2a). On average, excretion rates increased by 48% across the observed range of temperature (23.0–30.2 °C) (mean and 95% CI: $\beta = 0.05$ [0.01, 0.10]) (Fig. 2b) and by 100% across the observed range of AR (0.76–3.30) (square-root transformed; mean and 95% CI: $\beta = 0.71$ [0.25, 1.17]) (Fig. 2c). Moreover, the average temperature coefficient ($Q_{10}$; i.e., the relative change in carbonate excretion rate for every 10 °C rise in temperature) across the observed temperature range was 1.74 (95% CI: 1.06, 2.73). Carbonate excretion rate was negatively related to RIL (natural-log transformed) (Fig. 1a). This relationship was described by a power function with an

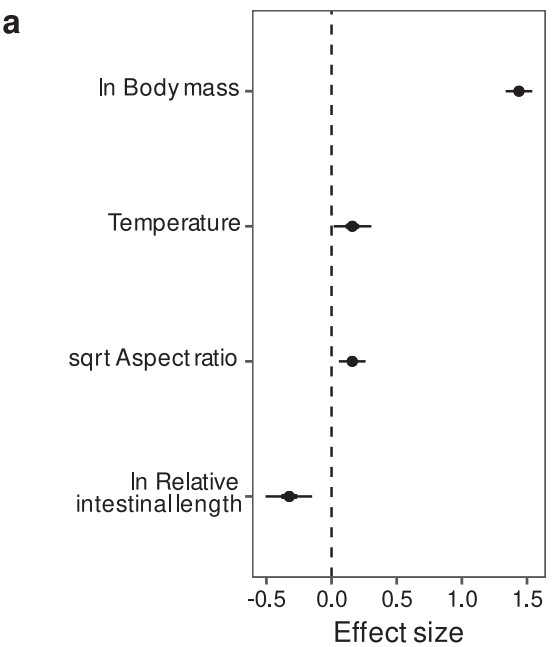

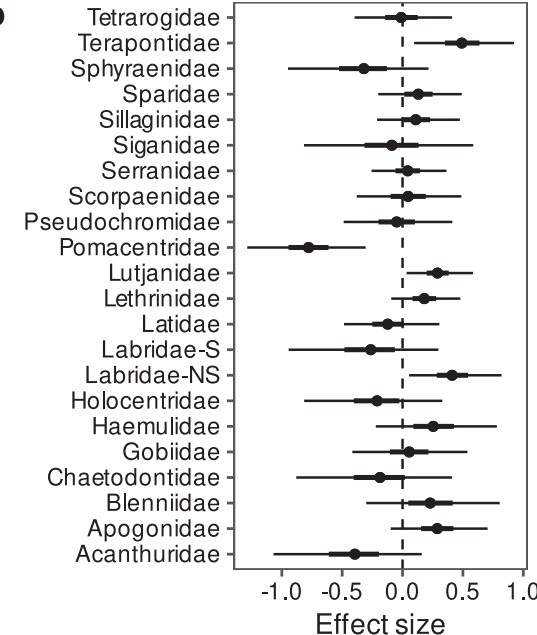

**Fig. 1 | Predictors of reef fish carbonate excretion rate. a** Effects of fish traits and temperature on carbonate excretion rate. **b** Family-specific effects on carbonate excretion rate. Estimates are medians (circles), 50% credible intervals (CIs; thick lines; some are too narrow to be seen) and 95% CIs (thin lines) derived from 12,000 posterior draws of a Bayesian multilevel distributional regression model. All predictors were standardised (mean-centred and scaled by one standard deviation) prior to fitting the model to allow for the comparison of effect sizes (non-standardised effects are reported in the text). Labridae-S scarine Labridae, Labridae-NS non-scarine Labridae. Data underlying the figures are available in the Zenodo repository (https://doi.org/10.5281/zenodo.7530455)[93].

exponent of −0.59 (95% CI: −0.92, −0.27), which translates into an average 82% decrease in excretion rate across the observed RIL range (0.33–7.56) (Fig. 2d).

### Predictors of carbonate composition

The identified predictors of carbonate excretion rate (i.e., body mass, AR, RIL, temperature, and family) were used to predict the excretion rate of five major carbonate polymorphs produced by fishes (i.e., LMC, HMC, aragonite, MHC, ACMC) using a Bayesian multivariate hurdle-lognormal model. The hurdle model allowed us to account for the large zero-inflation in the responses (14–83% of zeros) by modelling the probability of excretion of each polymorph as a function of RIL, temperature, and taxonomic identity, i.e., the variables known or expected to influence the mineralogical composition of excreted carbonates[28,38] (see the "Statistical modelling" section). The model predicted the correct proportion of zeros for all carbonate polymorphs (Supplementary Fig. 2) and showed a high predictive performance for positive observations, with a strong relationship between observed and predicted excretion rate for each polymorph (Supplementary Fig. 3). Further, over 96% of the observed values of each polymorph fell within the 90% CIs of the predictions (64–93% when considering the 50% CIs).

Fish body mass was consistently the strongest predictor of excretion rate for all carbonate polymorphs (Fig. 3a). The excretion rate of HMC scaled hypoallometrically with body mass (mean and 95% CI: $\beta = 0.74$ [0.68, 0.80]), whereas the excretion rate of MHC scaled hyperallometrically (mean and 95% CI: $\beta = 1.40$ [1.04, 1.78]), and that of other polymorphs did not differ from isometry as the wide uncertainty around the estimate overlapped with 1 (mean and 95% CI: $\beta = 0.89$ [0.56, 1.19], $\beta = 1.14$ [0.84, 1.39], $\beta = 0.89$ [0.64, 1.14], for LMC, aragonite, and ACMC, respectively). A positive effect of temperature and AR on excretion rate was consistent among polymorphs. Conversely, RIL negatively affected the excretion rate of ACMC and HMC (in agreement with the effect on total carbonate excretion rate), but had the opposite effect on the excretion rate of MHC and aragonite.

Temperature and RIL had relevant effects on the probability of excreting certain carbonate polymorphs (Fig. 3b). Temperature, for instance, positively influenced the probability of excreting ACMC, and to a lesser extent MHC and aragonite, and negatively affected the likelihood of fish excreting HMC. RIL was positively associated with the probability of excreting HMC and negatively associated with the likelihood of fish excreting aragonite and LMC.

For both aragonite and HMC, RIL had a contrasting effect on the excretion rate and probability of excretion (i.e., on the two parts of the hurdle model), with opposite patterns. Fish with longer intestines were less likely to excrete aragonite but excreted it at a higher rate than fish with shorter intestines. Conversely, fish with longer intestines were more likely to excrete HMC but excreted it at a lower rate compared to fish with shorter intestines. These contrasting effects resulted in right-skewed unimodal relationships between the excretion rate of the two polymorphs and RIL (Fig. 4), with the highest average excretion rates for aragonite and HMC in fish with RIL of about 1 and 1.4, respectively.

Although RIL and temperature increased the probability of excreting HMC and ACMC, respectively, carbonate composition was strongly conserved at the family level (Supplementary Figs. 4, 5). Indeed, most families showed large effect sizes on the probability of excreting certain polymorphs (Supplementary Fig. 4a). Nevertheless, a few families showed smaller effect sizes (e.g., Acanthuridae, Gobiidae), indicating higher intra-familial variability in carbonate composition. A weaker effect of family was observed on the excretion rate of a given polymorph (Supplementary Fig. 4b).

Furthermore, the multivariate model allowed us to estimate the correlations among the probabilities of excretion of different polymorphs after accounting for the effects of RIL and temperature (i.e., the group-level effect correlation). Specifically, we estimated correlations among polymorphs at the family level (Fig. 5). We found that families that were most likely to excrete HMC were less likely to excrete MHC, aragonite, or LMC. Conversely, the probabilities of excreting aragonite, LMC, and MHC were all positively correlated, highlighting that these polymorphs are generally co-produced by

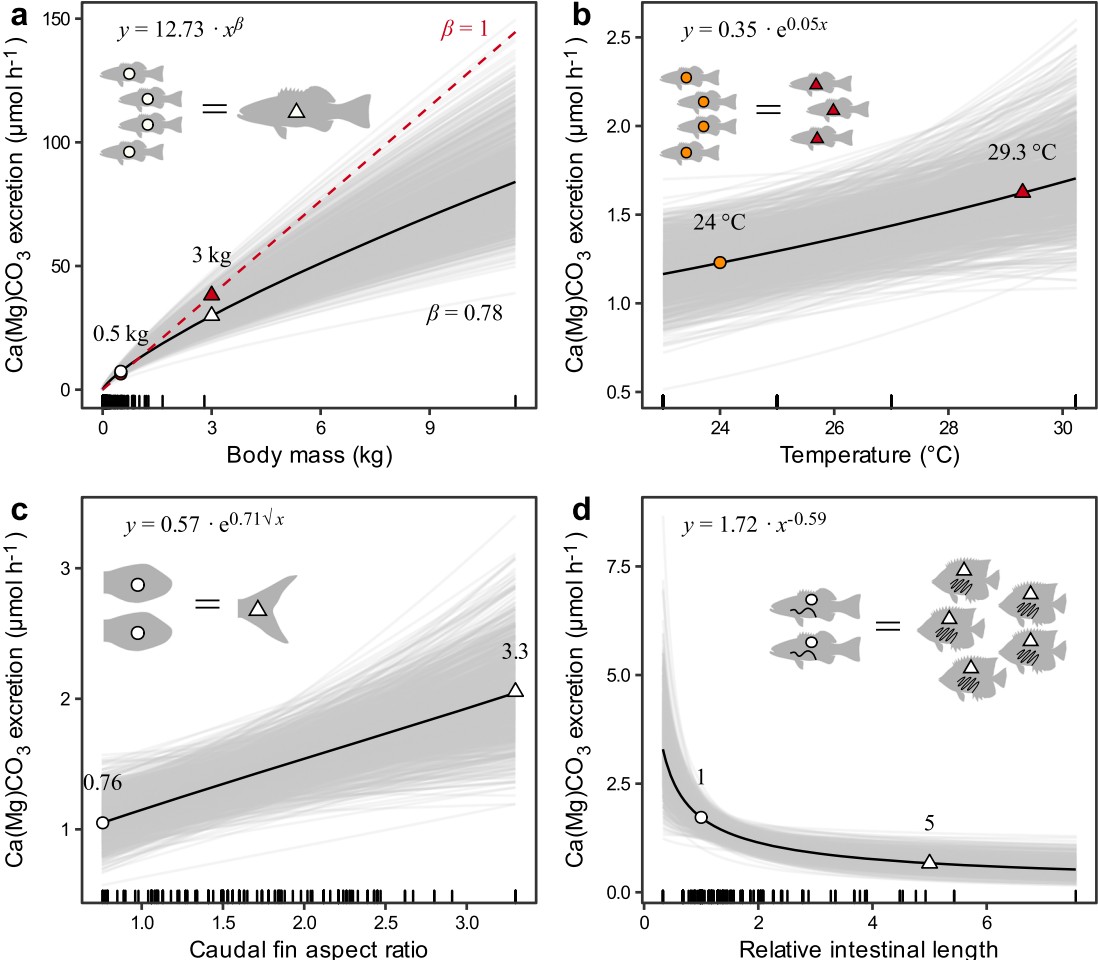

**Fig. 2 | Relationships between reef fish carbonate excretion rates, fish traits, and temperature.** Marginal effect of (**a**) body mass, (**b**) water temperature, (**c**) caudal fin aspect ratio (AR), and (**d**) relative intestinal length (RIL) after controlling for the remaining fixed and group-level effects of a Bayesian multilevel distributional regression model by standardising the other predictors at their mean values. Thick, black lines represent the mean predicted fits, whereas thin, grey lines represent 1000 draws randomly chosen from the posterior fits and show model fit uncertainty. Model predictions are for natural-log transformed excretion rates, but here show the fitted functions on the original scale of the data. Raw data are displayed as marks along the x-axis. In (**a**), the red dashed line represents isometric scaling ($\beta = 1$). In each panel, silhouettes show the number of fish required to excrete the same amount of carbonate at two levels of the predictor variable, which are also shown as matching symbols on the mean predicted fits. All silhouettes were drawn by MG and based on photographs taken by J.E. Randall and sourced from FishBase[78]. Data underlying the figures are available in the Zenodo repository (https://doi.org/10.5281/zenodo.7530455)[93].

fishes. ACMC may be excreted alongside all other polymorphs, although the probability was highest when LMC was also excreted.

## Discussion

Accurately assessing the role of fishes in the carbon cycle of the ocean requires a comprehensive understanding of the drivers of fish carbonate excretion rate and composition. Initial models of carbonate production in marine fishes primarily assumed a direct link to metabolic rate[8]. We demonstrate the relationships between fish carbonate excretion rate and three key drivers of metabolic rate (i.e., body mass, temperature, and AR), which show that the metabolism-carbonate excretion rate link is consistent across 71 reef fish species from 21 families. Furthermore, we show that this link is also mediated by RIL. These insights have important implications for quantifying community-level estimates of carbonate excretion rates and the indirect impacts of anthropogenic factors (mainly fishing and warming) on the contribution of fishes to the marine carbon cycle. Additionally, we provide evidence that carbonate excretion rate is related to body mass, temperature, and AR, and thus likely to fish metabolism, regardless of the carbonate polymorph excreted. However, intriguingly, polymorph-specific excretion rates differ in their relationship

with RIL. Finally, we show that the mineralogical composition of fish carbonates is highly conserved within families and to a lesser extent controlled by RIL and temperature. These findings allow refined estimates of carbonate excretion and composition at the regional and global scales to be generated. These estimates can be integrated into ocean carbonate and sediment production budgets and used in management and decision-making processes oriented towards the conservation of ecosystem functions[46].

Our multi-species analysis reveals that carbonate excretion rate scales hypoallometrically with body mass, as does metabolic rate. The estimated average scaling exponent of 0.78 is in agreement with the value of 0.80 found for resting metabolic rate across fishes[47] and ectotherms[48]. This suggests that carbonate excretion is directly proportional to metabolic rate through the effect of body mass. Furthermore, due to the scaling exponent <1, size-selective fishing and warming[49,50] will increase carbonate excretion rate per unit biomass by reducing fish size, thus averting immediate functional collapse as biomass is depleted[51,52] and large fish are extirpated[53].

The observed positive relationships of carbonate excretion rate with temperature and AR also support a direct link of carbonate excretion with metabolism. We found a $Q_{10}$ of 1.74, which is lower than

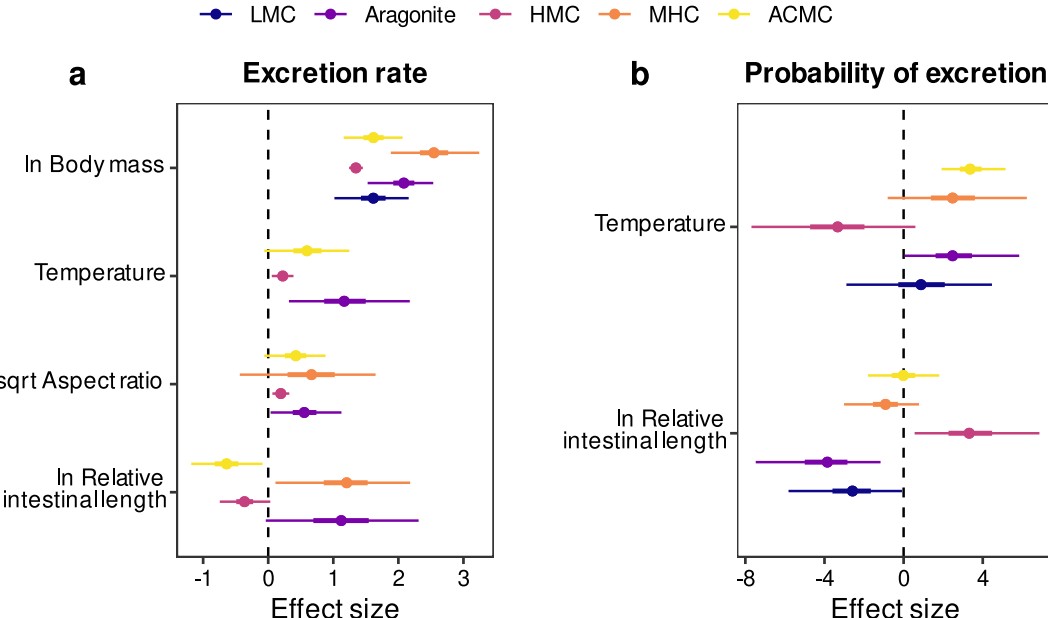

**Fig. 3 | Predictors of reef fish carbonate composition. a** Effects of fish traits and temperature on the excretion rate of five different carbonate polymorphs. **b** Effects of species relative intestinal length (RIL) and water temperature on the probability of excreting five different carbonate polymorphs. Estimates are medians (circles), 50% credible intervals (CIs; thick lines; some are too narrow to be seen) and 95% CIs (thin lines) derived from 6000 posterior draws of a Bayesian multivariate hurdle-lognormal model. All predictors were standardised (mean-centred and scaled by

one standard deviation) prior to fitting the model to allow for the comparison of effect sizes. Missing estimates correspond to effects excluded from the final model (see the "Statistical modelling" section). LMC Low-magnesium calcite, HMC High-magnesium calcite, MHC Monohydrocalcite, ACMC Amorphous calcium magnesium carbonate. Data underlying the figures are available in the Zenodo repository (https://doi.org/10.5281/zenodo.7530455)[93].

previously-observed species-specific values[8,29]. This is consistent with observations for fish resting metabolic rate, where species-specific $Q_{10}$ values are highly variable and >2 on average[47], while observed values across species are typically < 2[47,54,55]. Clarke and Johnston[47] found a $Q_{10}$ of 1.83, calculated over a 0–30 °C temperature range and across 69 fish species. Recalculated over the range of temperature observed in our study, that value is adjusted to 1.75, very close to our estimated value. This suggests that resting metabolic rate would increase by 50% over the same temperature range in which carbonate excretion rate is predicted to increase by 48%. Similarly, the relationship between carbonate excretion and AR is in agreement with results linking this morphological trait to metabolic rate[55]. Indeed, if we were to extrapolate carbonate excretion rate across the broader range of AR analysed by Killen et al.[55] (i.e., 0.66–7.2), which includes pelagic fishes, we would obtain a 314% increase in excretion, which is roughly equivalent to the estimated 3.4-fold difference in resting metabolic rate. While other morphological traits are related to metabolic rate (e.g., gill surface area and muscle protein content)[55,56], they are also directly related to AR and linked to fish lifestyle[55]. Therefore, although we did not directly incorporate these traits in our models, they are accounted for by including AR.

Altogether, these results support the prior assumption that carbonate excretion rate is directly proportional to metabolic rate and therefore support previous global estimates[8]. Our findings also suggest that the observed relationships could be extended outside the range of body mass and AR considered here, including large and pelagic fishes for which data collection is constrained by space availability in most research stations. However, we show that RIL has also a strong negative effect on carbonate excretion rate, thus affecting its direct link with metabolic rate. Consequently, the model used by Wilson et al.[8] appears to generally overestimate carbonate excretion rate for fishes with a RIL > 1 (Fig. 6). Their model was indeed parameterised using data from two benthic, predatory fishes with RIL

typically < 1[57,58], which produced estimates comparable to those of our model for similar fishes (e.g., AR = 1.5 and RIL = 0.5), regardless of temperature (Fig. 6 and Supplementary Fig. 6). Furthermore, a constant (i.e., $\rho$) was added to the earlier model to account for the higher resting metabolic rate of most fishes living in the water column and thus provide more realistic estimates[8]. According to our model, this correction is comparable to a seven-fold difference in AR (e.g., from 0.5 to 3.5), and leads to overestimates of the excretion rate for the large majority of fish species (Fig. 6). Therefore, carbonate production by marine fishes (at least at rest) may be lower than previously estimated. Furthermore, as RIL is negatively related to trophic level[59,60], the negative power relationship between excretion rate and RIL suggests that fishing and climate-induced regime shifts may reduce fish carbonate excretion by decreasing the mean trophic level of fish communities, thus potentially counteracting the buffering effect triggered by size-selective fishing. The effect of RIL on the individual carbonate polymorphs also suggests that these impacts may lead to shifts in community-level carbonate composition.

The negative relationship between carbonate excretion rate and RIL is counterintuitive from a metabolic point of view. With the intestine being an energetically expensive organ[61], metabolic and carbonate excretion rates should increase with increasing RIL. However, the energetic investment in the intestine may be balanced by the size of other expensive organs[62]. Factors unrelated to metabolism may thus explain this result. In our dataset, RIL was negatively related to salinity ($r = -0.60$), suggesting that part of the RIL effect may be confounded with the positive salinity effect on carbonate excretion[25,30,63]. However, the salinity range in our study (33.8–36.6) is an order of magnitude lower than the range driving a -2.5 fold change in carbonate excretion rate in the Gulf toadfish[30]. Therefore, salinity should explain minimal, if any, variability in our carbonate excretion rates.

Fishes with long intestines have a large intestinal surface area[60] and long gut residence time[64,65], which may enhance water absorption

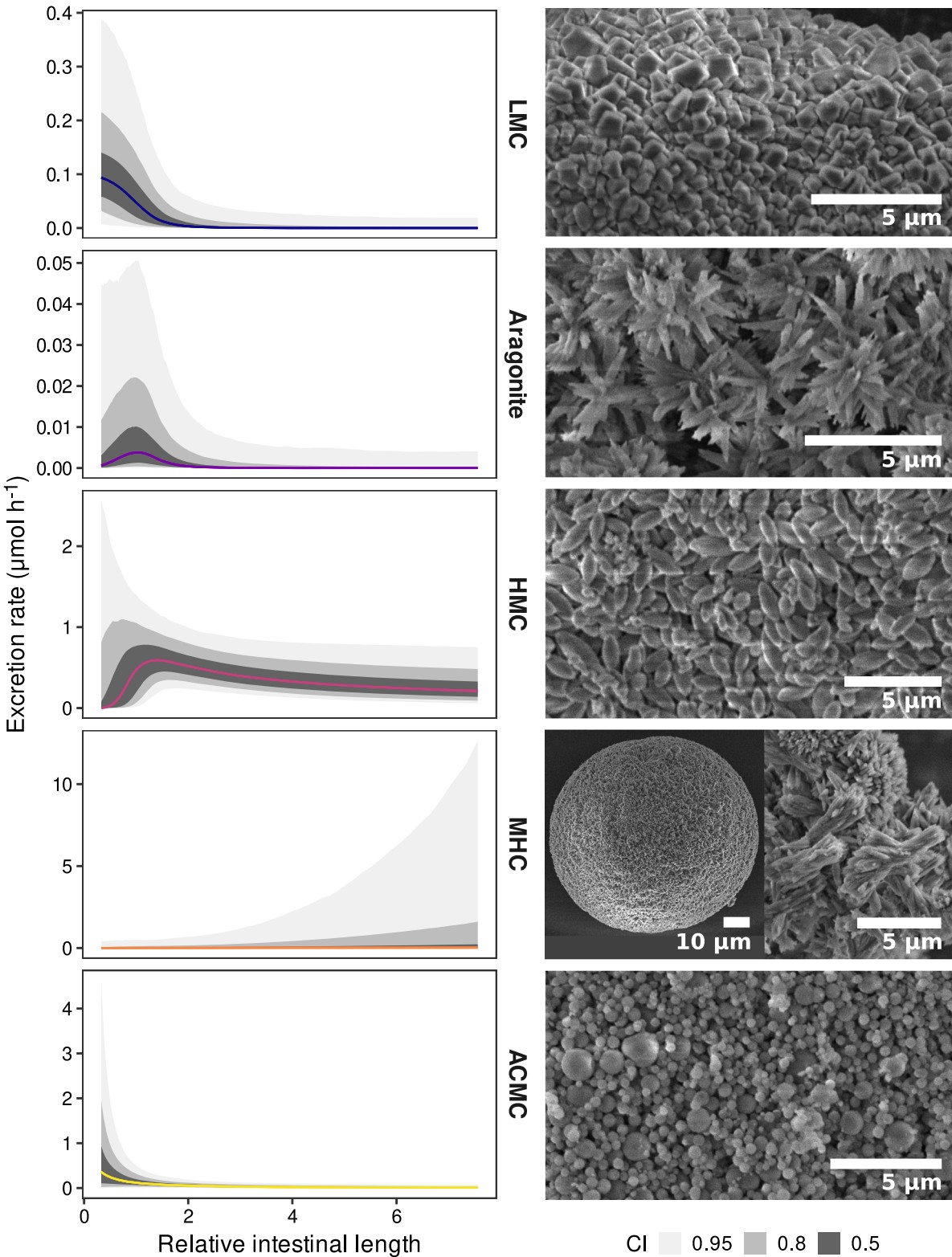

**Fig. 4 | Relative intestinal length affects reef fish carbonate composition.**
Marginal effect of species relative intestinal length (RIL) on the excretion rate of five different carbonate polymorphs after controlling for the remaining fixed and group-level effects of a Bayesian multivariate hurdle-lognormal model by standardising the other predictors at their mean values. Coloured lines represent the median predicted fits and the ribbons show the 50%, 80%, and 95% credible intervals (CI) around the estimate. Note the different scales on the y-axis. Scanning electron microscope images showing representative crystalline morphologies for each carbonate polymorphs are displayed on the right of each plot. LMC Low-magnesium calcite, HMC High-magnesium calcite, MHC Monohydrocalcite, ACMC Amorphous calcium magnesium carbonate. Data underlying the figures are available in the Zenodo repository (https://doi.org/10.5281/zenodo.7530455)[93].

efficiency. These fishes would presumably require correspondingly lower drinking rates and excrete lower amounts of carbonate. Water absorption efficiency has been measured in fishes with relatively short intestines[18–20,25,66–70]. To the best of our knowledge, no values of water absorption efficiency are currently available for fishes with RIL > 2. Nevertheless, the known range of water absorption efficiency (38.5–85%[22]) suggests that fishes with the lowest absorption efficiency must drink more than twice as much seawater as fishes with the highest absorption efficiency, with a direct effect on carbonate excretion. We also hypothesise that long gut residence times directly reduce

carbonate excretion rates, while potentially leading to accumulation of carbonate and irregular or delayed release of larger pellets, as previously reported for three temperate species[38]. Thus, we cannot discard that some of the excretion rates measured here could be affected by the relatively short sampling period considered (median: 64 h; range: 18–169 h). Simultaneous measurements of RIL, water absorption efficiency, and gut residence times are needed to better understand the mechanistic link underlying the observed relationships.

Residence time may also prove a viable explanation of the observed effect of RIL on carbonate composition. In this context, a longer residence time may allow unstable ACMC to transform into more stable polymorphs, such as aragonite and calcite, or into metastable MHC, which may then undergo further transformation[71–73]. We show that ACMC excretion rate is highest in fishes with very short intestines, while aragonite and HMC excretion rates are highest in fishes with intermediate RIL, and MHC is mostly excreted by fishes with long intestines. These results are consistent with previous observations that synthetic ACMC requires much longer time to transform into MHC than into aragonite and calcite[73]. Available compositional data from this and previous studies[28,37] for fish families not included in the analysis (due to low sample size) also confirm the observed patterns, with ACMC being the major polymorph produced by families with short intestines (e.g., Muraenidae) and MHC by families with long intestines (e.g., Zanclidae). Such mechanisms deserve further investigation and an analysis of carbonate development through the length of the intestine of species producing different polymorphs would permit testing of this hypothesis.

Nevertheless, RIL is highly phylogenetically conserved[60,74], leading to strong conservatism in carbonate composition at the family level. Our findings reiterate recent observations that the mineralogical composition of fish carbonates is broadly consistent within families across regions[28] and over large thermal gradients[38]. The family Labridae (excluding Scarini), however, has been highlighted as an exception to this general pattern in that it produced mainly ACMC with minor calcite in warm conditions (25–27 °C), but the opposite occurred at 10 °C[38]. A potential thermal control on the excretion of ACMC over calcite widely applicable across families has been recognised[38].

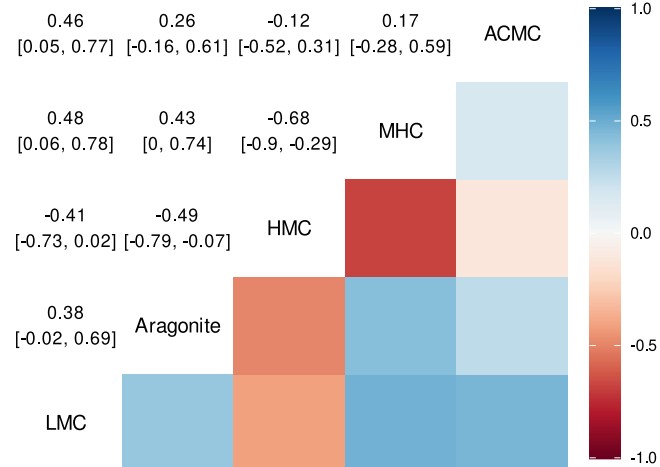

**Fig. 5 | Family-level correlations among the probabilities of excretion of five carbonate polymorphs by reef fishes.** Estimates are posterior medians and 95% credible intervals of correlation coefficients derived from 6,000 posterior draws of a Bayesian multivariate hurdle-lognormal model after controlling for temperature and relative intestinal length (Eq. 9). LMC, low-magnesium calcite; HMC, high-magnesium calcite; MHC, monohydrocalcite; ACMC, amorphous calcium magnesium carbonate. Data underlying the figures are available in the Zenodo repository (https://doi.org/10.5281/zenodo.7530455)[93].

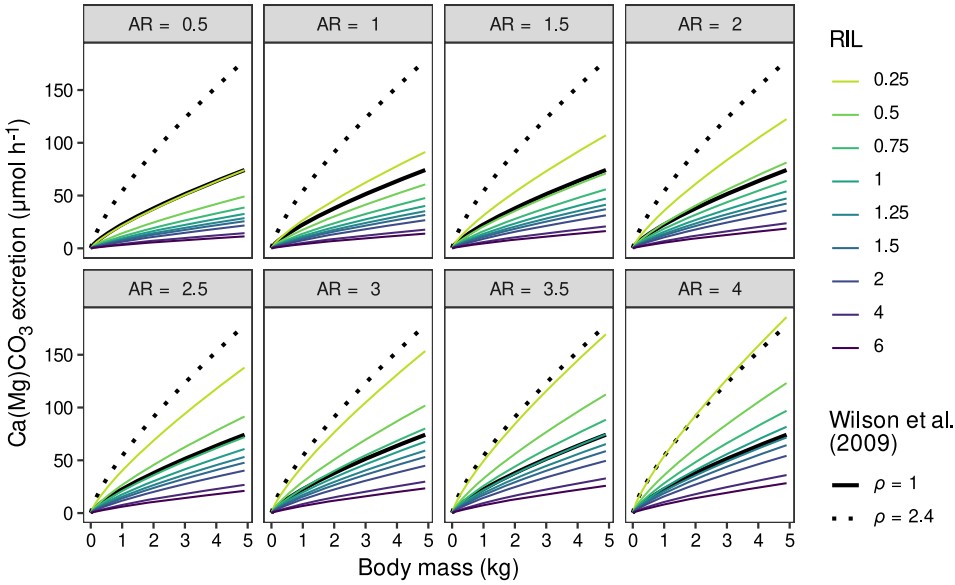

**Fig. 6 | Comparison between the estimates of our model and those of the model used by Wilson et al.[8].** Estimates of our model are average predictions (and thus do not account for the effect of family) at different levels of caudal fin aspect ratio (AR) and relative intestinal length (RIL). Estimates of "Wilson's model" are presented at two levels of the constant $\rho$ ($\rho = 1$ corresponds to benthic, sedentary fishes, whereas $\rho = 2.4$ is the value used to adjust model estimates for pelagic fishes with faster

resting metabolic rates), while the constant $\alpha$ was set to 1 to be comparable to our estimates (see Wilson et al.[8] for further details). All estimates are presented for a fixed temperature of 25 °C as results are unchanged at different temperatures (see Supplementary Fig. 6). Data underlying the figures are available in the Zenodo repository (https://doi.org/10.5281/zenodo.7530455)[93].

Our results provide evidence in support of this control given that we find a strong positive association between temperature and the probability of fishes excreting ACMC across families. Regardless of the underlying mechanism (increased gut residence times at lower temperature[75] have been suggested[38]), carbonates excreted by fish in warmer temperatures would contain more ACMC and less HMC (Supplementary Fig. 7), suggesting higher solubility and a reduced export of carbonate particles into the deep sea[12,35]. The associated reduced removal of alkalinity from surface waters weakens ocean acidification and favours the $CO_2$ uptake of the ocean. However, higher carbonate dissolution also lowers the $CO_2$ uptake by the biological carbon pump. This pump increases the $CO_2$ uptake of the ocean via the fixation of $CO_2$ into biomass (photosynthesis) and its export from the oceanic surface towards the deep sea, which is accelerated by carbonate minerals by increasing the density of sinking particles[76,77]. Hence, our findings have strong implications for understanding the role of fish carbonates in the marine carbon cycle and sequestration.

Our models provide substantial improvements to previous carbonate production models based on the parsimonious relationship with body mass and temperature[8,27,28,37] and allow us to produce species-level estimates under given thermal conditions. A major improvement lies in the ability to directly predict species-level excretion rates for individual carbonate polymorphs. These advances come, however, with limitations that have to be considered for any future application of our results and models. While variation in carbonate excretion rates among families is relatively small, carbonate composition is strongly conserved at the family level, hampering extrapolation to unsampled families. As the taxonomic scope of the existing carbonate database remains limited (35 reef-associated fish families out of 158[78], with 14 having < 3 observations), a targeted data collection campaign (preferably including small pelagic fishes) is needed to increase the proportion of fish biomass for which carbonate excretion rate can be predicted. Nevertheless, our models allow us to predict carbonate excretion rate and composition for several of the most abundant and biomass-rich fish families on coral reefs. Furthermore, the models were trained with data collected in shallow tropical and subtropical reefs, thereby limiting the potential geographic scope of their application. Additional data are needed to broaden the applicability of our results to marginal, non-reef, high-latitude, and pelagic environments. Progress has been made in expanding existing compositional data to temperate regions[38]. If associated production data becomes available, a potential extension of our models to temperate regions is possible.

It should also be noted that our models predict carbonate excretion rates for fishes that are most likely close to their resting metabolic rate, as data were collected from fasting fishes held individually (or in small monospecific groups) in relatively small tanks. However, as we found that carbonate excretion rate is likely directly related to metabolic rate, excretion rates of free-living fishes should scale proportionally to their metabolic rate. Previous studies have applied a common scaling factor to all species to overcome this issue[8,27,28,37,51]. This value was derived from a study which estimated the factorial activity scope (i.e., the ratio of field metabolic rate to resting metabolic rate[79]) for the Atlantic cod[80]. It is thus unlikely that this estimate is representative of all species, particularly for tropical reef fishes. Current knowledge on the factorial activity scope of reef fishes is limited and this parameter is likely highly variable among taxa and across body size[79,81]. The use of a single value as scaling factor may thus potentially introduce large errors in the estimated carbonate excretion rates. Instead, we suggest that future studies provide estimates of carbonate excretion rate for fishes at rest and discuss this limitation.

Improvements in the measurement of field metabolic rates in fishes[81,82] may soon allow updates to our models to predict carbonate excretion rates for fishes in their natural habitat. To do so, we must, however, consider the role of fish diet in carbonate production.

Fish diet is often rich in calcium, leading to high luminal calcium concentration with a direct effect on carbonate precipitation[23,63]. It would therefore seem reasonable to expect carbonate excretion rates several times higher in feeding compared to fasting fishes. It is thus likely that calcium (and in minor measure magnesium) obtained from food accounts for a large proportion of the carbonate excreted in wild feeding fishes. Diet may also alter intestinal fluid composition, and thus the precipitated carbonate polymorphs. However, existing data show no difference in the composition of carbonates excreted by fishes when feeding or fasting[37]. The calculation of calcium ingestion rates and subsequent carbonate precipitation given food ingestion rates[16] and food calcium content may be a feasible way to greatly improve carbonate production models.

## Methods

Animal collection and holding for this project was conducted under Marine Research Permit RE-19-28 issued by the Ministry of Natural Resources, Environment, and Tourism of the Republic of Palau (10.03.2019), Marine Research/Collection Permit and Agreement 62 issued by the Koror State Government (08.10.2019), Queensland Government GBRMPA Marine Parks Permit G14/36689.1, Queensland Government DNPRSR Marine Parks Permits QS2014/MAN247 and QS2014/MAN247a, Queensland Government General Fisheries Permit 168991, Queensland Government DAFF Animal Ethics approval CA2013/11/733, approval by The Bahamas Department of Marine Resources, approval by the Animal Care Officer of both the University of Bremen and the Leibniz Centre for Tropical Marine Research (ZMT), and in accordance with UK and Germany animal care guidelines.

### Sample collection

We collected fish carbonate samples at four study locations across three tropical and subtropical regions: Eleuthera (24°50′N, 76°20′W), The Bahamas, between 2009 and 2011[27,37]; Heron Reef (23°27′S, 151°55′E) and Moreton Bay (27°29′S, 153°24′E) in Queensland, Australia, in 2014 and 2015[28]; and Koror (7°20′N, 134°28′E), Palau, during November and December 2019. These are located within four distinct marine biogeographic provinces and three realms (Tropical Atlantic, Central Indo-Pacific, and Temperate Australasia)[43]. At each location fish were collected using barrier nets, dip nets, clove oil or hook and line, and immediately transferred to aquaria facilities at the Cape Eleuthera Institute, Heron Island and Moreton Bay Research Stations, and the Palau International Coral Reef Center. Fish were held in a range of tanks (60, 400, or 1400 L in the Bahamas, 10, 60, 100, 120, or 400 L in Heron Island and Moreton Bay, and 8, 80, 280, or 400 L in Palau) of suitable dimensions for different fish sizes (<1 g to 11 kg), either individually or, for particularly social species, in small groups of similar sized individuals of the same species. All tanks were supplied with flow-through locally-drawn filtered (1–5 μm) natural seawater, except in Moreton Bay where we used locally-drawn filtered natural seawater in a recirculation system, and maintained at ambient conditions. Food was withheld throughout the sampling period (typically three days but sometimes shorter or longer) and for at least 48 h prior to ensure that sample material comprised only carbonate precipitated within the intestine from imbibed seawater calcium, rather than from dietary sources. Additionally, each tank was fitted with a false mesh bottom to prevent further disturbance of excreted carbonate pellets or potential ingestion by fish. Carbonate pellets were collected from the bottom of the tanks using a siphon or disposable Pasteur pipette at 24 h intervals in The Bahamas and Australia (except a few non-scarine Labridae that were sampled at 4 h intervals—see ref. [28]), and at 8 h intervals in Palau. Samples were rinsed three times with deionised water (centrifuging each time for 3 min at 2655 x g) to remove saltwater and excess salts and soaked in sodium hypochlorite (commercial bleach; < 4% available chlorine) for 6–12 h to disaggregate organic material[83]. All traces of bleach were removed with further rinses with deionised water before

drying the samples for 24 h at 50 °C. Full details of carbonate collection in The Bahamas and Australia are described in refs. [27,28,37].

### Sample analysis

**Carbonate composition.** Samples were characterised for their morphological and mineralogical composition using scanning electron microscopy (SEM), energy-dispersive X-ray spectroscopy (EDX), and Fourier-Transform Infrared spectroscopy (FTIR). Detailed procedures for samples collected in The Bahamas and Australia are described in refs. [28,37]. The same methodology was applied to samples collected in Palau. Morphological and chemical composition were analysed using a Tescan Vega 3 XMU SEM with integrated Oxford Instruments X-MAX EDX detector. Dry samples were mounted on adhesive carbon tape and covered with a 20 nm conductive coating (Au). Morphological observations were made on at least five pellets per sample and electron microscope images were acquired at accelerating voltages between 5 and 15 kEV and working distances of 7–12 mm using either secondary electron or backscatter detectors. A minimum of 30 EDX scans were performed on each sample, incorporating all present particle morphotypes, using an accelerating voltage of 20 keV, a working distance of 15 mm, and acquisition time of > 40 s. Scans were only performed on particles surrounded by others of similar morphology, or on particles of sufficient size to ensure that data were representative of a particular morphology.

Mineralogical composition was assessed using Attenuated Total Reflection FTIR (ATR-FTIR). Spectra were obtained by the co-addition of 32 repeated scans performed at a resolution of 2 cm$^{-1}$ using a Nicolet 380 FTIR spectrometer coupled with a Thermo Scientific SMART iTR ATR sampler equipped with a diamond reflecting cell. To ensure spectra were representative, analyses were performed on at least three sub-samples (each comprising 2–3 pellets) per species. Spectra were then compared against an extensive spectral database for the identification of carbonate phases (see ref. [37]).

Finally, each particle morphotype produced by a species was assigned to a carbonate polymorph based on compositional and mineralogical data. The relative abundances of different particle morphotypes were then estimated visually for every sample observed using SEM. These were converted into relative abundances of carbonate polymorphs, which were then averaged for each species.

**Carbonate excretion rates.** The amount of carbonate excreted by fish in The Bahamas and Australia was quantified using a double-titration approach[18,27,28]. Samples were homogenised in 20 mL of distilled water, titrated with HCl to below pH 4.0 and titrated back to the starting pH with NaOH, while continuously aerating with $CO_2$-free air to remove all $HCO_3^-$ and $CO_3^{2-}$ as gaseous $CO_2$. Concentrations of 0.001–0.1 N were used for both HCl and NaOH as appropriate for sample size. Titrations were performed using a Metrohm Titrando autotitrator and Metrohm Aquatrode pH electrode (Australian samples), or manually with combination pH electrodes (Radiometer PHC 2401) and handheld pH meters (Hanna HI 8314 and Russell RL 200), with acid and base delivery via 2 mL micrometer syringes (Gilmont Instruments, Barrington, USA) with a precision of ±1 µL (Bahamian samples). The amount of HCl used minus the amount of NaOH required to return to the starting pH corresponds to the amount of bicarbonate equivalents (i.e., $HCO_3^- + 2CO_3^{2-}$) in the sample. Therefore, the molar amount of (Ca, Mg)$CO_3$ in the sample was calculated as:

$$n\left[(Ca,Mg)CO_3\right] = 0.5 \cdot n\left(HCO_3^-\right) = 0.5 \cdot (n(HCl) - n(NaOH)), \quad (1)$$

assuming that each carbonate molecule yields two bicarbonate equivalents.

Due to laboratory constraints, a slightly different approach was applied to the samples collected in Palau. Specifically, carbonate alkalinity was determined by single end point titration using the mixed indicator Bromocresol Green-Methyl Red[84]. Samples were suspended in 5 mL of distilled water and sonicated, then 50 µl of mixed indicator was added to the solution which turned blue (pH > 5). Each sample was titrated with 0.01–0.5 N HCl (with continuous aeration with $CO_2$-free air) until the end point (grey-lavender; pH–4.80) was reached and stable for at least 10 min. If the sample was over-titrated (pink), 0.01–0.1 N NaOH was added to titrate back to the end point and the amount of base used was subtracted from the amount of acid. Acid and base were added using an electronic multi-dispenser pipette (Eppendorf Repeater ®E3X, Eppendorf, Hamburg, Germany) with a precision of ±1 µL. Additionally, the pH of several samples was monitored using a pH microelectrode (Mettler Toledo InLab Micro) to ascertain the correctness of the colorimetric end point. The amount of carbonate in the sample was then calculated using Eq. (1). The method was validated using certified reference material (Alkalinity Standard Solution, 25,000 mg/L as CaCO3, HACH) and the accuracy in the determination of solid samples was verified using certified CaCO3 powder (Suprapur, ≥ 99.95% purity, Merck) samples (60–500 µg) and resulted in 96.53 ± 1.94% accuracy (mean ± SE; $n = 8$).

To compare values obtained with the two titration methods we further analysed 12 samples collected at Lizard Island, Australia, in February 2016. Samples were collected at 24 h intervals from one individual of *Lethrinus atkinsoni* (f. Lethrinidae, body mass: 245 g), a group of five *Lutjanus fulvus* (f. Lutjanidae, mean body mass: 21 g), and an individual of *Cephalopholis cyanostigma* (f. Serranidae, body mass: 295 g), following the procedures described above. During sample collection water temperature ranged from 29.1 °C during the night to 32.6 °C during the day, with an average of ~31 °C, mean salinity was 35.4, and pH$_{NBS}$ ranged from 8.13 to 8.21. To compare the amount of carbonate measured by the two methods we added carbonate samples to 20 ml ultrapure water and disaggregated crystals via sonication. We then used a Metrohm Titrando autotitrator and Metrohm Aquatrode pH electrode to measure initial pH of the suspension of carbonates, then titrated each sample of carbonate in two stages. Firstly, they were titrated down to pH 4.80 using 0.1 M HCl, adding 20 µl increments of acid until this was sufficient to keep pH below 4.80 for 10 min whilst bubbling with $CO_2$-free air. This first stage was comparable to the single end point titration used for samples collected in Palau. Secondly, whilst continuing to bubble with $CO_2$-free air, further acid was added to the sample until it reached pH 3.89 and was stable for 1 min. Then 0.1 M NaOH was added to the samples to return them to the initial pH. For all samples the first end point titration (to pH 4.80) yielded slightly higher values for carbonate content than the second double titration. The ratio between the two methods (single end point/ double titration) was 1.08 ± 0.01 (mean ± SE; range: 1.04–1.14; Supplementary Table 2). As we found a small but consistent difference between the two methods, all following analyses were initially performed on the actual data obtained with the double titration for samples from Australia and The Bahamas, and the single end point titration for samples from Palau. Then, to assess the robustness of the results, we repeated the analyses after applying a correction factor of 1.08 to the excretion rates of Palauan fishes (that used the single end point titration method). All results were consistent and robust to the measured difference between the titration methods (Supplementary Figs. 8, 9).

Finally, measurements of multiple samples from each individual collected over periods of 18–169 h (median: 64 h) were combined to produce an average individual excretion rate in µmol h$^{-1}$. For fish held in groups, carbonate excretion rates per individual (of average biomass) were obtained by averaging the total excretion rate of the group across the sampling period and dividing it by the number of individuals in the tank. Excretion rates obtained from fish groups thus evened the intraspecific variability within tanks, and are therefore more robust than those directly obtained from fish held individually. This aspect

was considered in our models by fitting weighted regressions (see the "Statistical modelling" section). In total, we measured the carbonate excretion rates of 382 individual fishes arranged in 192 groups (i.e., independent observations), representing 85 species from 35 families across three tropical regions (180 individuals from 29 species in Australia, 90 individuals from 10 species in the Bahamas, and 112 individuals from 46 species in Palau; Supplementary Table 1).

We assume that during the sampling of carbonates fishes were close to their resting metabolic rate and that their carbonate excretion rates are representative of fish at rest. Although the ratio of tank volume to fish volume in our study (median ~660; inter-quartile range ~180–1700) typically greatly exceeds the guideline ideal range for measuring resting metabolic rate (20–50)[85], fishes were fasted prior to and throughout sampling, and in most instances their movement was somewhat constrained by tank volume. Fasting reduces metabolic rate in all animals, including fish, as they do not undergo energy-intensive digestive processes and use energy reserves to support vital processes, triggering metabolic changes in many tissues and reducing activity levels[86,87]. Additionally, other than the carbonate syphoning (< 2 min), no stressors were present. Fishes were not engaged in foraging activity, they experienced no predator-prey interactions, many were held individually so did not engage in social interactions, and social species were held in groups to minimise stress. Therefore, although spontaneous activity likely occurred, fishes were placid throughout the sampling period and the constrained space, minimal disturbance, and fasting suggest that they had very low activity levels.

### Selection of families

To assess the main determinants of variability in fish carbonate excretion and composition we considered only families with at least three independent observations (i.e., three individuals or groups of fish). This removed 14 families characterised by only one or two observations, thus reducing our dataset to 175 independent observations from 352 individuals representing 21 families. In our analyses, we considered parrotfishes (f. Labridae: tribe Scarini) separately from other labrids following ref. [28], as preliminary data showed that they excrete distinct carbonate products, possibly due to their distinct trophic ecology and unique gut chemistry[88].

### Explanatory variables

Fish carbonate production is the result of both extrinsic environmental conditions and intrinsic species traits. To analyse the major factors determining carbonate excretion rate and composition we considered a suite of potential variables, while accounting for taxonomic relationships. Some are known to influence fish carbonate excretion rates, such as salinity, body mass, and temperature[8,25,27,63]. Others, such as $CO_2$ and AR, are likely to indirectly affect carbonate excretion by influencing acid–base regulation[31] and activity level[55], respectively. We did not have data for seawater $p\mathrm{CO_2}$ during the sampling period, however, due to good aeration of tanks, we assumed that it was close to atmospheric equilibrium (~400 µatm) at all locations, and thus would not have been a relevant factor in our analysis.

Furthermore, diet is expected to strongly influence carbonate excretion rate and composition, as fish obtain large amounts of calcium and magnesium from their food, which are then likely largely precipitated and excreted as carbonates[23]. However, our data were collected on fasting fishes, thus we cannot account for the direct effect of diet on carbonate excretion and composition in our analyses. Nevertheless, we accounted for the indirect effect of diet (i.e., the adaptation of the gut morphology to the typical diets each species has evolved to consume) by including RIL as an additional potential variable in our analyses. Indeed, RIL is strictly related to diet in reef fishes[60]. As we did not have direct measurements of intestinal length of fishes used to collect carbonate samples, we predicted species-level RIL using a Bayesian phylogenetic model trained with the largest

available dataset of intestinal length of reef fishes[89] (see Supplementary Methods).

### Statistical modelling

**Predictors of total carbonate excretion.** Before modelling carbonate excretion rates, we used bivariate correlations to identify potential multicollinearity among all explanatory variables, including two covariates related to our methodologies (total sampling period and titration method). The titration method was strongly correlated with all environmental variables as one protocol was applied to all samples from Palau and the other to all samples from Australia and The Bahamas. Titration method was therefore initially excluded from our models. Conversely, salinity was relatively strongly correlated with RIL ($r = -0.60$) in our dataset. Therefore, these variables were alternatively included in our models (i.e., the same models were fitted twice including either RIL or salinity as a covariate).

We fitted a series of Bayesian regression models to predict carbonate excretion rates based on the selected traits and environmental variables. Let $y_{ijk}$ be the carbonate excretion rate of the $i^{th}$ individual of the $j^{th}$ species, belonging to the $k^{th}$ family. We assumed that each observation of the response variable ($y_{ijk}$) was $t$-distributed:

$$
\begin{aligned}
y_{ijk} &\sim \mathrm{t}\left(\nu, \mu_{ijk}, \sigma\right) \\
\sigma &\sim \mathrm{t}(3, 0, 2.5) \\
\nu &\sim \Gamma(2, 0.1)
\end{aligned}
\tag{2}
$$

with degrees of freedom $\nu$, scale $\sigma$, and observation-specific locations $\mu_{ijk}$ defined as

$$
\begin{aligned}
\ln\left(\mu_{ijk}\right) &= \beta_{0k} + \beta_x x \\
\beta_{0k} &= \gamma_0 + u_k \\
u_k &\sim \mathrm{N}\left(0, \tau_{u_k}\right) \\
\gamma_0, \beta_x, \tau_{u_k} &\sim \mathrm{N}(0, 5)
\end{aligned}
\tag{3}
$$

where $\gamma_0$ is the average model intercept, $u_k$ is the random variation in $\gamma_0$ based on taxonomic family, and $\beta_x$ is a vector of regression coefficients of the fixed effects $x$.

We fitted a series of 36 linear and multilevel models starting from an intercept-only model and increasing in complexity. All models were fitted by weighting the response variable based on whether fish were kept individually or in groups. Although some fish were kept in relatively large groups (up to 13 individuals), most were kept individually (61% of tanks). Therefore, to avoid overweighting observations from groups larger than two individuals, we gave a weight of two to all observations derived from fish kept in groups of two or more individuals. We built linear models by first including body mass which was the known major predictor. We then added either RIL or salinity, which in our exploratory data analysis showed the strongest correlation with the response after accounting for body mass. Lastly we included temperature, AR, and total sampling period, either alone or in combination. This procedure resulted in 18 linear models which were then refitted including taxonomic family as a group-level effect. Model selection was performed through leave-one-out cross-validation (LOO-CV) (Supplementary Table 3). All multilevel models had a better fit than the corresponding linear models highlighting the importance of including the fish family as a group-level effect. Similarly, all models including RIL performed better than the same models where RIL was replaced by salinity. The selected model included the following set of covariates:

$$
\beta_x x = \beta_1 \ln(M)_{ijk} + \beta_2 \ln(RIL)_{jk} + \beta_3 \sqrt{AR_{jk}} + \beta_4 T_{ijk}
\tag{4}
$$

where $M$ is the body mass (in kg), $RIL$ and $AR$ are the species-level relative intestinal length and caudal fin aspect ratio (the latter obtained from FishBase[78]), respectively, and $T$ is the average water temperature (in °C) during the sampling period.

To investigate whether there was some unexplained variance in the response that could be attributed to the excluded explanatory variables, we tested for correlations between model residuals and average salinity during the sampling period, titration method, and total sampling period. No residual correlation was observed. However, residual variance was related to the titration method used to quantify carbonate excretion rate (Levene's test, $F_{1,173} = 22.82$, $p < 0.001$), with a larger residual variance in samples analysed through single end point titration. Therefore, to account for this, we refitted the selected model as a distributional regression where we allowed the scale parameter $\sigma$ of the $t$-distribution to vary with respect to the titration method used:

$$
\begin{aligned}
\ln\left(\sigma_{ijk}\right) &= \gamma_\sigma + \beta_\sigma \text{method}_{ijk} \\
\gamma_\sigma &\sim t(3, 0, 2.5) \\
\beta_\sigma &\sim N(0, 5)
\end{aligned}
\tag{5}
$$

where $\gamma_\sigma$ is the intercept and $\beta_\sigma$ is the regression coefficient for the titration method. Thus, $\sigma = \exp(\gamma_\sigma)$ for observations obtained through the reference method (i.e., double titration), while $\sigma = \exp(\gamma_\sigma + \beta_\sigma)$ for observations obtained through single end point titration.

Model comparison through LOO-CV showed that modelling $\sigma$ as a function of the titration method improved model fit (Supplementary Table 4). Moreover, the distributional model showed that the parameter $\sigma$ was actually different between methods (mean and 95% CI: 0.39 [0.31, 0.47] and 0.80 [0.63, 0.98], for double and single end point titration, respectively). This model was therefore selected to draw conclusions on the relationship between carbonate excretion rate and the explanatory variables.

**Predictors of carbonate composition.** Five major carbonate polymorphs are produced by fish: LMC, aragonite, HMC, MHC, and ACMC, in order of increasing expected solubility. To investigate the factors determining the excretion of the different carbonate minerals by fish we modelled the excretion rates of individual polymorphs. This approach has two major strengths: (1) it facilitates the investigation of both what drives the probability of a polymorph being excreted, and the predictors of the polymorph-specific excretion rates, and (2) it allows direct predictions of polymorph-specific excretion rates.

To obtain the excretion rate of individual polymorphs, the carbonate excretion rate of each fish was multiplied by the species-level relative carbonate composition. Then, we modelled these excretion rates using a Bayesian multivariate hurdle-lognormal model. We used a multivariate model (i.e., a model with multiple response variables) because it accounts for the correlation among polymorphs within taxonomic group, while allowing the use of different sets of predictors for each response. As all response variables contained zeros (14 to 83% of the observations), we opted for a hurdle-lognormal model, which is a two-part model that combines a logistic regression for the probability that the outcome is zero or not, with a lognormal model for the non-zero responses:

$$
\Pr(y|\theta, \mu, \sigma) = \begin{cases} \theta & \text{if } y = 0, \text{ and} \\ (1 - \theta)\frac{\log N_{(y|\mu,\sigma)}}{1 - \log N_{CDF}(0|\mu,\sigma)} & \text{if } y > 0, \end{cases}
\tag{6}
$$

where $\theta$ is the probability of zero outcome (i.e., no excretion), $(1 - \theta)$ is the probability of positive outcome (i.e., excretion), and $logN_{CDF}$ is the cumulative distribution function for the lognormal distribution of the non-hurdle part.

The hurdle probability of each carbonate polymorph ($\theta_{ijk}^m$), i.e., the probability that the $i^{th}$ individual of the $j^{th}$ species, belonging to the $k^{th}$ family, did not excrete the $m^{th}$ polymorph, was estimated using a multilevel logistic regression as:

$$
\begin{aligned}
\text{logit}\left(\theta_{ijk}^m\right) &= \beta_{0k}^m + \beta_1^m \ln(\text{RIL})_{jk} + \beta_2^m T_{ijk} \\
\beta_{0k}^m &= \gamma_0^m + \upsilon_k^m \\
\gamma_0^m &\sim \text{logistic}(0, 1) \\
\beta_{1:2}^m &\sim N(0, 5)
\end{aligned}
\tag{7}
$$

Where $\gamma_0^m$ is the average intercept for the $m^{th}$ polymorph, $\upsilon_k^m$ is the random variation in $\gamma_0^m$ based on taxonomic family, and $\beta_{1:2}^m$ are the regression coefficients of RIL and T, respectively. While, for each response, we modelled the mean of the lognormal distribution ($\mu_{ijk}^m$) according to Eqs. (3) and (4) and the standard deviation ($\sigma_{ijk}^m$) according to Eq. (5). Body mass and AR were only included as predictors of the excretion rates, but not as predictors of the probability of excretion of different carbonate polymorphs, because no mechanistic link is described or expected for these variables. Conversely, we used fish family, temperature, and RIL to predict the probability of excretion of the polymorphs because fish carbonate mineralogy is generally consistent within families[28,38], a potential thermal effect has been suggested[38], and fish with long intestines have long gut residence times[64,65] likely affecting the precipitation of different polymorphs.

To account for both the between-family variance ($\tau^2$) and covariance ($\rho$) we modelled the group-level effects as correlated. Therefore, we assumed the family-specific intercepts of both the hurdle ($\upsilon_k^m$) and non-hurdle part ($u_k^m$) of the model to follow a multivariate normal distribution with zero means and covariance matrix $\Sigma$ with $2m(2m + 1)/2$ components:

$$
\begin{bmatrix} u_k^1 \\ u_k^2 \\ \vdots \\ u_k^m \\ \upsilon_k^1 \\ \upsilon_k^2 \\ \vdots \\ \upsilon_k^m \end{bmatrix} \sim N\left(\mu = \begin{bmatrix} 0 \\ 0 \\ \vdots \\ 0 \\ 0 \\ 0 \\ \vdots \\ 0 \end{bmatrix}, \sum\right)
\tag{8}
$$

$$
\sum = \begin{bmatrix}
\tau_{u_k^1}^2 & \rho\tau_{u_k^1}\tau_{u_k^2} & \cdots & \rho\tau_{u_k^1}\tau_{u_k^m} & \rho\tau_{u_k^1}\tau_{\upsilon_k^1} & \rho\tau_{u_k^1}\tau_{\upsilon_k^2} & \cdots & \rho\tau_{u_k^1}\tau_{\upsilon_k^m} \\
\rho\tau_{u_k^2}\tau_{u_k^1} & \tau_{u_k^2}^2 & \cdots & \rho\tau_{u_k^2}\tau_{u_k^m} & \rho\tau_{u_k^2}\tau_{\upsilon_k^1} & \rho\tau_{u_k^2}\tau_{\upsilon_k^2} & \cdots & \rho\tau_{u_k^2}\tau_{\upsilon_k^m} \\
\vdots & \vdots & \ddots & \vdots & \vdots & \vdots & \ddots & \vdots \\
\rho\tau_{u_k^m}\tau_{u_k^1} & \rho\tau_{u_k^m}\tau_{u_k^2} & \cdots & \tau_{u_k^m}^2 & \rho\tau_{u_k^m}\tau_{\upsilon_k^1} & \rho\tau_{u_k^m}\tau_{\upsilon_k^2} & \cdots & \rho\tau_{u_k^m}\tau_{\upsilon_k^m} \\
\rho\tau_{\upsilon_k^1}\tau_{u_k^1} & \rho\tau_{\upsilon_k^1}\tau_{u_k^2} & \cdots & \rho\tau_{\upsilon_k^1}\tau_{u_k^m} & \tau_{\upsilon_k^1}^2 & \rho\tau_{\upsilon_k^1}\tau_{\upsilon_k^2} & \cdots & \rho\tau_{\upsilon_k^1}\tau_{\upsilon_k^m} \\
\rho\tau_{\upsilon_k^2}\tau_{u_k^1} & \rho\tau_{\upsilon_k^2}\tau_{u_k^2} & \cdots & \rho\tau_{\upsilon_k^2}\tau_{u_k^m} & \rho\tau_{\upsilon_k^2}\tau_{\upsilon_k^1} & \tau_{\upsilon_k^2}^2 & \cdots & \rho\tau_{\upsilon_k^2}\tau_{\upsilon_k^m} \\
\vdots & \vdots & \ddots & \vdots & \vdots & \vdots & \ddots & \vdots \\
\rho\tau_{\upsilon_k^m}\tau_{u_k^1} & \rho\tau_{\upsilon_k^m}\tau_{u_k^2} & \cdots & \rho\tau_{\upsilon_k^m}\tau_{u_k^m} & \rho\tau_{\upsilon_k^m}\tau_{\upsilon_k^1} & \rho\tau_{\upsilon_k^m}\tau_{\upsilon_k^2} & \cdots & \tau_{\upsilon_k^m}^2
\end{bmatrix}
\tag{9}
$$

with

$$
\begin{aligned}
\tau_{u_k^m} &\sim N(0, 5) \\
\tau_{\upsilon_k^m} &\sim N(0, 5) \\
\rho &\sim \text{LKJCorr}(1)
\end{aligned}
\tag{10}
$$

Finally, we fitted a second model specifying a different formula for the mean of the lognormal distribution of each polymorph ($\mu_{ijk}^m$). This was achieved by removing the fixed effects with relatively large errors

(i.e., those with an estimated error greater than the mean estimate). Specifically, we removed the effects of RIL, AR, and temperature on LMC and the effect of temperature on MHC. Model comparison through LOO-CV showed no difference in model fit between the two models, therefore, we selected the more parsimonious model to create figures and interpret results.

All analyses were performed with the software program R (version 4.1.3[90]) and all models were fitted with the R package *brms* (version 2.15.0[91]). Linear and multilevel models were run for 4 chains, each with 4000 iterations and a warm-up of 1000 iterations, whereas hurdle models were run for 3 chains, each with 4000 iterations and a warm-up of 2000 iterations. All models were examined for evidence of convergence using trace plots and Gelman–Rubin statistics and we used posterior predictive distributions to check for models' fit.

### Reporting summary
Further information on research design is available in the Nature Portfolio Reporting Summary linked to this article.

## Data availability
The data generated and/or analysed in this study have been deposited in the Zenodo repository (https://doi.org/10.5281/zenodo.7530092)[92]. The intestinal length data used in this study are freely available in the Zenodo repository (https://doi.org/10.5281/zenodo.5172790)[89]. Data underlying all figures in the main text and Supplementary Information are available in the Zenodo repository (https://doi.org/10.5281/zenodo.7530455)[93].

## Code availability
The code to reproduce all analyses and figures is available on GitHub (https://github.com/mattiaghilardi/FishCaCO3Model) and Zenodo (https://doi.org/10.5281/zenodo.7530092[92]).

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

## Acknowledgements
We thank the staff at the Palau International Coral Reef Centre (PICRC), Cape Eleuthera Institute, Heron Island and Moreton Bay Research Stations for support during fieldwork, S. Flotow for his help with SEM-EDX analysis, and P. Lewin, S. Bröhl, and E. Fossile for their help during data collection in Palau. This work was primarily funded through the 2017–2018 Belmont Forum and BiodivERsA REEF-FUTURES project under the BiodivScen ERA-Net COFUND program (awarded to DM) through the French National Research Agency (ANR-18-EBI4-0005) (DM and VP) and the DFG (BE6700/1-1) (SB). Data collection in The Bahamas and Australia and sample analysis was funded by the UK Natural Environment Research Council (NERC) grants NE/K003143/1 (CTP and RWW), NE/G010617/1 (CTP and RWW), and NE/H010041/1 (RWW), and a Biotechnology and Biological Sciences Research Council (BBSRC) grant BB/J00913X/1 (RWW). We wish to thank the two reviewers whose thoughtful input significantly improved the manuscript.

## Author contributions
M.G. and S.B. conceptualised the study. M.G., M.A.S., C.T.P., R.W.W., M.B., A.B., and S.B. collected the data. M.G. analysed the data and led the writing of the paper. M.A.S., V.P., S.C.A.F., T.R., C.W., M.B., C.T.P., A.B., R.W.W., D.M., and S.B. contributed significantly to the drafts and approved the final version.

## Funding

## Competing interests
The authors declare no competing interests.
