## [Peer Review File · Nature Communications]

Temperature, species identity and morphological traits predict carbonate excretion and mineralogy in tropical reef fishesREVIEWER COMMENTS

Reviewer #1 (Remarks to the Author):

This study used measurements of carbonate excretion rate and carbonate polymorph type determined from 71 species (representing 21 families) of coral reef fish to test whether different fish traits, primarily those related to metabolism, can predict these carbonate attributes.

A handful of previous studies have determined carbonate excretion rates and carbonate mineral composition in a wide range of fishes (including previous work conducted by several of the co-authors). While their conclusions suggest fishes can contribute significantly to carbonate production and flux and potentially ocean alkalinity, these studies do not provide enough information to determine robust mechanistic relationships between carbonate excretion and mineralogy/morphology and traits that control them. This is a similar problem as those measuring other forms of carbon released from fish (e.g., fecal carbon, excreted DOC), and therefore we still cannot really constrain fish-based carbon flux or even provide empirically derived biomass-specific information for carbon/biogeochemical model integration. As such, defining those relationships and identifying the primary drivers of carbon flux have recently been identified as high priority research needs (see for example Saba et al. 2021 cited within the manuscript) so that we can begin to determine potential impacts of climate change and/or fishing on fish-based carbon contributions to the biological pump and importantly, in this case, as a potential buffer for ocean acidification.

The present study uses empirical data and a Bayesian approach to provide this much needed confidence in the previously assumed predictability of fish traits and carbonate excretion rate (robust results show that carbonate excretion rate scales proportionally with metabolic rate, linked to body mass, temperature, and aspect ratio). This study also went a step further to investigate the factors driving the excretion of five major carbonate polymorphs, which have varying levels of dissolution rates – a characteristic that would drive how much alkalinity would be produced during the sinking of the carbonate particle through the water column. This study is, therefore, an important contribution to the specific field of fish physiology and the broader field of ocean carbon biogeochemistry and will move these respective fields forward with applicability to biogeochemical modeling parameterization and improvement.

General comments:

The study focuses on tropical and subtropical reef fishes, most likely residing in shallow coastal environments. How applicable are these predicted relationships to fishes inhabiting other environments (e.g., in offshore, deeper, colder waters) or to fishes with more active swimming behaviors (e.g., larger, migratory fishes). The authors state these three locations are “characterized by distinct climatic features”, but do not offer expansion on how they vary and how applicable this variability would be to offshore, non-reef environments. I guess the assumption is that if carbonate excretion is related to metabolism that these relationships would extend to other fishes, but there is limited discussion on this point specifically on the confidence in this application to fishes with different lifestyles, swimming mode, etc. The range of fish biomass in the empirical data was “<1 g to >10 kg”, but there seems to be only 1 empirical biomass greater than 3 kg (1 datapoint somewhere around 11 kg from Fig. 2a). Those are still small fish relative to the full range in the marine system (e.g., tuna = 250 kg), and also may not represent the full range of lifestyles and swimming modes which can affect metabolic rates and scaling of those rates with biomass. So authors should discuss these possibilities in more detail than that already included in the discussion.

The methods contain a fairly vague and minimal description on fish husbandry and experimental set-up for the empirical data collection. How long did the studies take place? In the “tanks of suitable dimensions” (line 432): 1) if fish were housed in groups

to collect carbonates, how did the authors estimate biomass-to-carbonate excretion rate ratios? Did they take a mean of individual biomass in each tank and associate that mean with their "average individual excretion rate"? if so, then am I correct to assume they didn't incorporate intraspecific biomass ranges in their study? (it's not clear in the text); and 2) were they small enough to minimize much movement, or were fish actively swimming? This is important because when authors use 'metabolism', are they referring to standard/basal metabolism (SMR) or active/maximum metabolic rates (MMR) that may scale differently with biomass (and certainly temperature)? They assume resting rates (lines 399-400), but state the fish are in tanks and not small chambers with specific biomass to surrounding water volume ratios as are traditionally used for determining resting metabolic rates. See earlier work by Douglas Glazier, Shaun Killen, and discussion in Norin & Clark 2015 J Fish Biol that MMR scales with an exponent higher than SMR. How did the authors address this in their models when incorporating biomass and temperature?

I feel like the authors missed an opportunity to discuss their results, particularly those focused on the carbonate polymorphs, in relation to buffering capacity for ocean acidification. These particles produce alkalinity as they dissolve, and do so while sinking through the water column. I think a short paragraph near the end of the discussion would be worthwhile to highlight the results of this study in that light and would attract a broader range of readers.

The 5 discussion paragraphs on potential reasons for the negative relationship between carbonate excretion rate and RIL could be cut down and made more concise. It seemed to be a bit unbalanced (in relative length, no pun intended) in the discussion, particularly because the RIL data used were not part of the empirical measurements but instead gathered from FishBase. Were the authors making the point that RIL should be a more common measurement to better pinpoint the mechanistic explanation for the negative relationship with carbonate excretion rate? And if so, I think it more efficient for the authors to directly state that.

Other minor comments:

Line 99: and subtropical?

Line 104: What are these distinct climatic features? and are they distinct enough to be representative of non-tropical fishes or fishes inhabiting open ocean waters?

Line 106: Supplemental doc says, 414 observations from 65 species from 24 families. Why is this different (and are observations = individuals)?

Line 273: How does higher individual-level carbonate excretion rate avert immediate functional collapse?

Lines 275-291: see above comment about SMR v. MMR. Their trends with temperature can vary greatly and that should be addressed here. Are the authors assuming their measured rates are fish at SMR (or MMR or somewhere in between)?

Reviewer #2 (Remarks to the Author):

The study from Ghilardi et al. analyzes the potential drivers of fish carbonate excretion rates and composition, looking at environmental factors, fish traits and fish families. The authors compiled data from tropical and subtropical reef fish and found that the carbonate excretion rate across these fish species is positively related to body mass, temperature and AR while being negatively impacted by RIL. Part of the results support the expected proportionality between carbonate excretion rate and the metabolic rate of fish but altogether the results suggest a deviation from this proportionality through the

negative influence of RIL. The mineralogical composition of carbonates excreted by fish is showed to be stable among fish families and their probability of excretion is also influenced by temperature and RIL.

This is a well designed study with an intensive statistical analysis of the results. The methods are sound and it will be an important addition to the existing literature on the role of fish to the carbon cycle. I only have minor comments to the authors.

Comments to the authors:

I31, I114 You mention that the excretion rate "decreases disproportionately with body mass" while later on I.143 for example and on Fig. 1a we see that the excretion rate is related positively to body mass through an hypo-allometric relationship. So do you mean the amount of carbonate per unit of mass decreases on lines 31 and 114 ? I find the wording misleading/confusing.

I170 "the identified predictors" can you specify ? This sentence suggests that some of the predictors are left out, are you referring to salinity or total sampling period ? I think you only mention earlier salinity as another tested predictor (others are in the methods), maybe clarify it here.

I174 Here it is not clear to me why only RIL, temperature and taxonomic identity are used to model the probability of excretion. Is it the best fit model already ?

I233 Can you specify the kind of correlation ?

I560 you mention that CO₂ can potentially affect carbonate excretion, do you know if the presence of anoxic zones in the gut of fish have a notable impact on carbonate excretion ? Can it be linked to RIL ?

Figure 5: I found difficult to rapidly understand this figure. I think this is because you more or less represent three times the same information (i.e. the correlations) using 1) the colors, 2) the circle sizes and 3) the values. In my opinion, this figure would be more easily readable by only representing the bottom triangle with the colors filling entire squares (and not changing circle sizes) and the values on top all written in black.

Response To Reviewers

ID: NCOMMS-22-28203

Title: Temperature, species identity, and morphological traits regulate carbonate excretion and mineralogy in tropical reef fishes

Reviewer 1 (Remarks to the Author):

This study used measurements of carbonate excretion rate and carbonate polymorph type determined from 71 species (representing 21 families) of coral reef fish to test whether different fish traits, primarily those related to metabolism, can predict these carbonate attributes.

A handful of previous studies have determined carbonate excretion rates and carbonate mineral composition in a wide range of fishes (including previous work conducted by several of the co-authors). While their conclusions suggest fishes can contribute significantly to carbonate production and flux and potentially ocean alkalinity, these studies do not provide enough information to determine robust mechanistic relationships between carbonate excretion and mineralogy/morphology and traits that control them. This is a similar problem as those measuring other forms of carbon released from fish (e.g., fecal carbon, excreted DOC), and therefore we still cannot really constrain fish-based carbon flux or even provide empirically derived biomass-specific information for carbon/biogeochemical model integration. As such, defining those relationships and identifying the primary drivers of carbon flux have recently been identified as high priority research needs (see for example Saba et al. 2021 cited within the manuscript) so that we can begin to determine potential impacts of climate change and/or fishing on fish-based carbon contributions to the biological pump and importantly, in this case, as a potential buffer for ocean acidification.

The present study uses empirical data and a Bayesian approach to provide this much needed confidence in the previously assumed predictability of fish traits and carbonate excretion rate (robust results show that carbonate excretion rate scales proportionally with metabolic rate, linked to body mass, temperature, and aspect ratio). This study also went a step further to investigate the factors driving the excretion of five major carbonate polymorphs, which have varying levels of dissolution rates – a characteristic that would drive how much alkalinity would be produced during the sinking of the carbonate particle through the water column. This study is, therefore, an important contribution to the specific field of fish physiology and the broader field of ocean carbon biogeochemistry and will move these respective fields forward with applicability to biogeochemical modeling parameterization and improvement.

Thank you for reviewing our manuscript and for highlighting its relevance and timely contribution. We appreciate the reviewer's suggestions, and we have made several changes throughout the manuscript to address the reviewer's comments.

General comments:

1. *The study focuses on tropical and subtropical reef fishes, most likely residing in shallow coastal environments. How applicable are these predicted relationships to fishes inhabiting other environments (e.g., in offshore, deeper, colder waters) or to fishes with more active swimming behaviors (e.g., larger, migratory fishes). The authors state these three locations are “characterized by distinct climatic features”, but do not offer expansion on how they vary and how applicable this variability would be to offshore, non-reef environments. I guess the assumption is that if carbonate excretion is related to metabolism that these relationships would extend to other fishes, but there is limited discussion on this point specifically on the confidence in this application to fishes with different lifestyles, swimming mode, etc. The range of fish biomass in the empirical data was “<1 g to >10 kg”, but there seems to be only 1 empirical biomass greater than 3 kg (1 datapoint somewhere around 11 kg from Fig. 2a). Those are still small fish relative to the full range in the marine system (e.g., tuna = 250 kg), and also may not represent the full range of lifestyles and swimming modes which can affect metabolic rates and scaling of those rates with biomass. So authors should discuss these possibilities in more detail than that already included in the discussion.*

We thank the reviewer for highlighting this aspect and agree that it should be further discussed due to its relevance in understanding the applicability of our results in the parameterisation of biogeochemical models. We previously mentioned that the potential geographic scope of the application of our models and results is limited as we focused on tropical and subtropical reef fishes, and that additional data from other regions, such as temperate areas, are required to extend their applicability (lines **422-428**). Nevertheless, as mentioned by the reviewer, we assume that, if carbonate excretion is proportionally related to metabolism, the observed relationship with temperature could be extended, at least to some extent. Similarly, the species considered in our study cover a wide range of body sizes, lifestyles, and trophic levels, but do not include large (except for a barracuda of ~11 kg) and/or pelagic fishes. However, previous estimates suggest that fish weighing <1 kg are responsible for the vast majority (>97%) of global fish carbonate production (Wilson et al. 2009, SI Table S1, <https://doi.org/10.1126/science.1157972>). So although our dataset excludes large fishes, the weight range it encompasses is probably a good representation of overall fish carbonate production. Furthermore, data collection from large fishes or active pelagic fishes is constrained by the research facilities available. Thus, extending the range of sizes and lifestyles in the carbonate database is less straightforward compared to extending the thermal range. Killen et al. (2016, <https://doi.org/10.1086/685893>) showed that pelagic fishes mainly differ from non-pelagics in their maximum metabolic rate (MMR), but only minimally in their standard/resting metabolic rate (SMR). As our data were collected from fish in conditions close to those used to measure SMR (see answer to comment 2 of reviewer 1) and we used a continuous predictor in our models, caudal fin aspect ratio, as proxy for general activity level and lifestyle, we are confident that the resulting relationship could be extended to active pelagic fishes with high caudal fin aspect ratio (>3). Indeed, our results are in agreement with those obtained by Killen et al. for SMR across a broader range of caudal fin aspect ratio (0.66-7.2), including pelagic fishes. The collection of additional data, perhaps from small pelagic fishes, would allow us to test this assumption. We have now incorporated these aspects in the discussion (lines **299, 306-309, 418** and **423-425**).

2. *The methods contain a fairly vague and minimal description on fish husbandry and experimental set-up for the empirical data collection. How long did the studies take place? In the “tanks of suitable dimensions” (line 432): 1) if fish were housed in groups to collect carbonates, how*

did the authors estimate biomass-to-carbonate excretion rate ratios? Did they take a mean of individual biomass in each tank and associate that mean with their “average individual excretion rate”? if so, then am I correct to assume they didn’t incorporate intraspecific biomass ranges in their study? (it’s not clear in the text); and 2) were they small enough to minimize much movement, or were fish actively swimming? This is important because when authors use ‘metabolism’, are they referring to standard/basal metabolism (SMR) or active/maximum metabolic rates (MMR) that may scale differently with biomass (and certainly temperature)? They assume resting rates (lines 399-400), but state the fish are in tanks and not small chambers with specific biomass to surrounding water volume ratios as are traditionally used for determining resting metabolic rates. See earlier work by Douglas Glazier, Shaun Killen, and discussion in Norin & Clark 2015 J Fish Biol that MMR scales with an exponent higher than SMR. How did the authors address this in their models when incorporating biomass and temperature?

We have now improved the description of fish husbandry and data collection. We have specified the size of the tanks used (lines **465-467**), sampling duration (line **471**), and the calculation of excretion rate for fish held in groups (lines **580-585**). In such cases it was not possible to determine the excretion rate of each individual in the tank as samples represented carbonate excreted by all fish. Therefore, we divided the average excretion rate of the group by the number of individuals in the tank to obtain an average excretion rate for an individual of average biomass (fish held in groups were of similar size to minimise variation). Although it was not possible to directly incorporate this source of intraspecific variation in our analysis, this was to some extent accounted for by using the average across individuals, and was considered in our models. Moreover, intraspecific variation in biomass is included for several species for which samples were collected from multiple individuals or groups (encompassing a range of body mass) held in separate tanks (Supplementary Table 1), although this was to some extent dictated by fish capture success.

Fishes of different body mass (<1 g to 11 kg) were held in tanks ranging between 8 and 1,400 L. Although they were not small chambers as used to determine resting metabolic rate, they were small enough to limit much movement as compared to fish in the wild. Furthermore, food was withheld for at least 48 h prior sampling and throughout the sampling period, and other than the carbonate syphoning (<2 min), no stressors were present. Fasting is known to reduce metabolic rate in all animals, including fish, as they do not undergo energy-intensive digestive processes and have to use energy reserves to support vital processes, triggering metabolic changes in many tissues and reducing activity levels (e.g., van Dijk et al. 2002, <https://doi.org/10.1007/s00442-001-0830-3>; Gingerich et al. 2010, <https://doi.org/10.1007/s00360-009-0419-4>). Therefore, we assume that during sampling fishes were close to their resting metabolic rate and that the measured carbonate excretion rates are representative of fish at rest. Our results appear to support this assumption. This has now been explained in the Methods section (lines **590-600**). We have also replaced the term “starved” with “fasting” throughout the manuscript as the former was incorrectly used.

3. *I feel like the authors missed an opportunity to discuss their results, particularly those focused on the carbonate polymorphs, in relation to buffering capacity for ocean acidification. These particles produce alkalinity as they dissolve, and do so while sinking through the water column. I think a short paragraph near the end of the discussion would be worthwhile to highlight the results of this study in that light and would attract a broader range of readers.*

We agree with the reviewer that this is an important aspect and we have now included a short discussion on the implication of our results for fish carbonate role in the buffering capacity for ocean acidification (lines **397-406**), supported by an additional figure (Supplementary Fig. 7).

4. *The 5 discussion paragraphs on potential reasons for the negative relationship between carbonate excretion rate and RIL could be cut down and made more concise. It seemed to be a bit unbalanced (in relative length, no pun intended) in the discussion, particularly because the RIL data used were not part of the empirical measurements but instead gathered from FishBase. Were the authors making the point that RIL should be a more common measurement to better pinpoint the mechanistic explanation for the negative relationship with carbonate excretion rate? And if so, I think it more efficient for the authors to directly state that.*

We agree with the reviewer that our previous discussion may have included disproportionate detail on the effect of RIL. We have now cut this part of the discussion (lines **338-371**). Nevertheless, we would like to highlight that, instead of the five paragraphs mentioned by the reviewer, only two paragraphs discussed potential reasons for the negative relationship between carbonate excretion rate and RIL and one between carbonate composition and RIL. This discussion shows that although RIL measurements are relatively common (data are available for numerous fish species), it should be measured together with water absorption efficiency and gut retention time to understand the mechanistic link underlying the observed relationship (lines **369-371**). Also, we would like to reiterate that RIL was not gathered from FishBase but predicted using a Bayesian phylogenetic model trained with empirical data, as explained in the Methods section (lines **652-654**) and Supplementary Methods.

Other minor comments:

5. *Line 99: and subtropical?*

Yes, now added to the sentence (line **105**).

6. *Line 104: What are these distinct climatic features? and are they distinct enough to be representative of non-tropical fishes or fishes inhabiting open ocean waters?*

Although there are different climatic conditions between the sampling locations, from hot and humid tropical conditions in Palau to sub-tropical with high seasonal variability in Moreton Bay, we realised that this statement is not the most appropriate here, and the conditions are not representative of higher-latitude or pelagic environments. We have now removed this statement and added the marine biogeographic provinces and realms to which the sampling locations belong (lines **111** and **460-462**).

7. *Line 106: Supplemental doc says, 414 observations from 65 species from 24 families. Why is this different (and are observations = individuals)?*

Line 106 referred to carbonate excretion rate data, whereas Supplementary Methods referred to a dataset of fish intestinal length, where each of the 414 observations was an individual. This dataset was only used to validate the extrapolation of intestinal length from a Bayesian phylogenetic model to unobserved

species (i.e., species not used to train the model). As this independent dataset is not yet published and still embargoed, it would not be freely accessible together with all other data. Therefore, we decided to follow the cross-validation approach described in Parravicini et al. 2020, <https://doi.org/10.1371/journal.pbio.3000702>), using the same data (including 1,208 individuals from 142 species and 31 families) for model training and validation, which allows us to have a fully open access publication where all data described in the manuscript are freely accessible. The extrapolation remains strongly supported by the new approach and this modification does not affect any results and conclusions of the manuscript. All changes are highlighted in blue in the new version of the Supplementary Information, including explanation of the new approach in paragraph two of the Supplementary Methods and results of the validation are now presented in Supplementary Fig. 10.

8. *Line 273: How does higher individual-level carbonate excretion rate avert immediate functional collapse?*

While fish biomass is depleted by size-selective fishing and warming, the smaller fish size and thus higher carbonate excretion rate per unit biomass results in a slower decrease in carbonate excretion than just based on total loss of biomass, averting immediate functional collapse. The mechanism has been described by Jennings and Wilson (2009, <https://doi.org/10.1111/j.1365-2664.2009.01682.x>) and is identical to that of fish biomass productivity explained by Morais et al. (2020, <https://doi.org/10.1111/gcb.14941>). This has now been clarified in the text (lines **283-286**).

9. *Lines 275-291: see above comment about SMR v. MMR. Their trends with temperature can vary greatly and that should be addressed here. Are the authors assuming their measured rates are fish at SMR (or MMR or somewhere in between)?*

We have now explained in the Methods section (lines **590-600**) our assumption that measured carbonate excretion rates are representative of fishes at rest. Therefore, we believe that the different scaling of SMR and MMR with temperature should not be addressed here.

Reviewer 2 (Remarks to the Author):

The study from Ghilardi et al. analyzes the potential drivers of fish carbonate excretion rates and composition, looking at environmental factors, fish traits and fish families. The authors compiled data from tropical and subtropical reef fish and found that the carbonate excretion rate across these fish species is positively related to body mass, temperature and AR while being negatively impacted by RIL. Part of the results support the expected proportionality between carbonate excretion rate and the metabolic rate of fish but altogether the results suggest a deviation from this proportionality through the negative influence of RIL. The mineralogical composition of carbonates excreted by fish is showed to be stable among fish families and their probability of excretion is also influenced by temperature and RIL.

This is a well designed study with an intensive statistical analysis of the results. The methods are sound and it will be an important addition to the existing literature on the role of fish to the carbon cycle. I only have minor comments to the authors.

Thank you for reviewing our manuscript and for the positive feedback.

Comments to the authors:

1. *131, 1114 You mention that the excretion rate “decreases disproportionately with body mass” while later on l.143 for example and on Fig. 1a we see that the excretion rate is related positively to body mass through an hypo-allometric relationship. So do you mean the amount of carbonate per unit of mass decreases on lines 31 and 114 ? I find the wording misleading/confusing.*

Thanks for pointing this out. The term is defined in line **65**. Nevertheless, we agree that it might be confusing and we have now rephrased lines 31 (now lines **34-35**) and 114 (now line **123**) to avoid confusion.

2. *1170 “the identified predictors” can you specify ? This sentence suggests that some of the predictors are left out, are you referring to salinity or total sampling period ? I think you only mention earlier salinity as another tested predictor (others are in the methods), maybe clarify it here.*

This has now been clarified here (now line **178**) and also in lines **116-118**.

3. *1174 Here it is not clear to me why only RIL, temperature and taxonomic identity are used to model the probability of excretion. Is it the best fit model already ?*

The reasons behind our selection of variables to model the probability of excretion have now been explained in lines **183-185** and lines **737-744**.

4. *1233 Can you specify the kind of correlation ?*

These are modelled correlations and the corresponding equation is referenced in the caption of Fig. 5. We have now specified that these are group-level effect correlation (line **245**), where the group is the fish family.

5. *1560 you mention that CO₂ can potentially affect carbonate excretion, do you know if the presence of anoxic zones in the gut of fish have a notable impact on carbonate excretion ? Can it be linked to RIL ?*

This is an interesting point. The potential effect of CO₂ on fish carbonate excretion is related to the equilibrium between pCO₂ in fish blood with pCO₂ in seawater. A pCO₂ increase in seawater would result in a corresponding increase in pCO₂ and secondary HCO₃⁻ in the blood. This would stimulate higher secretion of HCO₃⁻ (derived from both hydration of endogenous CO₂ and transepithelial transport of bicarbonate from

the blood) into the intestine, potentially leading to higher carbonate precipitation and excretion (Grosell 2019, <https://doi.org/10.1016/bs.fp.2019.07.002>). Therefore, we do not expect a notable effect of anoxic zones in the gut on carbonate precipitation and excretion, and we are not aware of any relationship between RIL and the presence of anoxic zones.

6. *Figure 5: I found difficult to rapidly understand this figure. I think this is because you more or less represent three times the same information (i.e. the correlations) using 1) the colors, 2) the circle sizes and 3) the values. In my opinion, this figure would be more easily readable by only representing the bottom triangle with the colors filling entire squares (and not changing circle sizes) and the values on top all written in black.*

We agree with the reviewer that figure 5 may not have been easily understood as the same information has been represented in multiple ways. We have now modified the figure following reviewer's suggestions.

REVIEWERS' COMMENTS

Reviewer #1 (Remarks to the Author):

The authors did a thorough job of addressing most of my previous comments, but I still would like some clarity regarding my comment about SMR and tank volume. I was referring to the guidelines described in Svendsen et al. 2016, doi:10.1111/jfb.12797, whereby tank volume-to-fish volume ratios should be between 20-50 to ensure organisms are comfortable but can still restrict movement and ensure a 10% drop in oxygen content or pO₂ within a brief time period. For example a 300 g fish would need a respirometry tank volume of less than ~15 L to meet these requirements. The tank volumes used in the excretion experiments in the authors' study were much higher, ranging from 60 to 1400 L, but they do not report the total fish biomass (or volume) in each tank. Although the authors were not directly measuring metabolic rates in this study, they are still making direct relationships/comparisons between carbonate excretion rates and resting metabolic rates that are measured using important specific guidelines to represent minimal activity, so I think it's important that there is as much clarity as possible.

Reviewer #2 had no more comments.

Response To Reviewers

ID: NCOMMS-22-28203A

Title: Temperature, species identity, and morphological traits regulate carbonate excretion and mineralogy in tropical reef fishes

Reviewer 1 (Remarks to the Author):

The authors did a thorough job of addressing most of my previous comments, but I still would like some clarity regarding my comment about SMR and tank volume. I was referring to the guidelines described in Svendsen et al. 2016, doi:10.1111/jfb.12797, whereby tank volume-to-fish volume ratios should be between 20-50 to ensure organisms are comfortable but can still restrict movement and ensure a 10% drop in oxygen content or pO₂ within a brief time period. For example a 300 g fish would need a respirometry tank volume of less than ~15 L to meet these requirements. The tank volumes used in the excretion experiments in the authors' study were much higher, ranging from 60 to 1400 L, but they do not report the total fish biomass (or volume) in each tank. Although the authors were not directly measuring metabolic rates in this study, they are still making direct relationships/comparisons between carbonate excretion rates and resting metabolic rates that are measured using important specific guidelines to represent minimal activity, so I think it's important that there is as much clarity as possible.

Thank you for reviewing our manuscript. We understand the reviewer's point and for clarity we have now included the tank volume-to-fish volume ratios in our study (median ~660; inter-quartile range ~180-1700) and expanded the explanation of our assumption that fishes were close to their resting metabolic rate during carbonate sampling (lines **520-537**). Although these values greatly exceed the guideline ideal range for measuring SMR (20-50) described in Svendsen et al. 2016a, doi:10.1111/jfb.12797, it should be noted that these guidelines are mainly constrained by technical issues to obtain accurate measurements of metabolic rate. Quoting Svendsen et al. 2016a, "Given an estimated range of oxygen consumptions, planning the respirometer volume often comes down to trying to achieve the best compromise between having a large enough drop in oxygen level during the measurement period to achieve a satisfactory r^2 , but doing it over a relatively short amount of time. While there are no hard and fast rules that can be applied in this regard, it has been the authors' experience that a respirometer:organism volume (r_{RO}) between 20 and 50 appears to be comfortable for most organisms but is small enough to result in a 10% drop in oxygen content or pO₂ (Forstner, 1983) within a reasonable amount of time (3–6 min), depending on temperature. The 10% drop in oxygen should not be regarded as a standard, as determinations with a high r^2 (>0.95–0.98) with a smaller decline in oxygen levels would suffice. On the other hand, if there are drops of 10% or more, all having low r^2 (<0.95), it should be a cause for concern.". Since we did not directly measure metabolic rates, these issues did not apply to our study and we could therefore use larger tanks which improve water quality and reduce fish stress.

Nevertheless, a larger tank volume-to-fish volume ratio decreases the precision of the SMR measurement, although it does not affect the mean SMR (Svendsen et al. 2016b, doi:10.1111/jfb.12851). In such cases, a longer measurement time is required to achieve a substantial enough decrease in oxygen content and a measurement with high r^2 and low variation. For instance, based on Figs 9 and 10 in Svendsen et al. 2016a, for our median ratio of tank volume to fish volume a measuring period of ~1-2 h would be required to reach a decline of 5% in oxygen content at our study temperatures. We sampled carbonates at 8 or 24 h intervals, longer than would be needed to measure SMR. While spontaneous activity likely occurred (as occurs within respirometry chambers), fishes were placid throughout the sampling period and activity was limited by fasting and reduced stress. Fishes were not engaged in foraging activity, they experienced no predator-prey interactions, many were held individually so did not engage in social interactions, and social species were held in groups to minimise stress. Testing in groups should also be considered when measuring metabolic rates in social species (Chabot et al. 2016, doi:10.1111/jfb.12845). Additionally, we sampled over multiple days and the measured carbonate excretion rates were averaged across the entire sampling period, thus minimising the influence of any potential bout in activity.

Reviewer 2 had no more comments.

Thank you for reviewing our manuscript.